# Casein Kinase 2 (CK2): A Possible Therapeutic Target in Acute Myeloid Leukemia

**DOI:** 10.3390/cancers15143711

**Published:** 2023-07-21

**Authors:** Øystein Bruserud, Håkon Reikvam

**Affiliations:** 1Institute for Clinical Science, Faculty of Medicine, University of Bergen, 5021 Bergen, Norway; hakon.reikvam@uib.no; 2Section for Hematology, Department of Medicine, Haukeland University Hospital, 5021 Bergen, Norway

**Keywords:** CK2, acute myeloid leukemia, prognosis, silmitasertib, chemotherapy, chemoresistance, proliferation, apoptosis, cytokine

## Abstract

**Simple Summary:**

Acute myeloid leukemia (AML) is an aggressive blood cancer disease that can only be cured by intensive anticancer treatment, and for several patients, allogeneic stem cell transplantation is needed. The protein kinase casein kinase 2 (CK2) is an intracellular signaling molecule that supports cellular growth and survival. This is true both for normal and cancer cells, and the growth- and survival-supporting effects seem to be more important for many cancer cells (including AML cells) than for normal cells. Several pharmacological inhibitors of CK2 have been developed, and this therapeutic strategy is now tried in patients with various malignant diseases. In this article, we review available studies suggesting CK2 inhibition as an effective strategy also for the treatment of AML, either as monotherapy or as a part of drug combinations.

**Abstract:**

The protein kinase CK2 (also known as casein kinase 2) is one of the main contributors to the human phosphoproteome. It is regarded as a possible therapeutic strategy in several malignant diseases, including acute myeloid leukemia (AML), which is an aggressive bone marrow malignancy. CK2 is an important regulator of intracellular signaling in AML cells, especially PI3K–Akt, Jak–Stat, NFκB, Wnt, and DNA repair signaling. High CK2 levels in AML cells at the first time of diagnosis are associated with decreased survival (i.e., increased risk of chemoresistant leukemia relapse) for patients receiving intensive and potentially curative antileukemic therapy. However, it is not known whether these high CK2 levels can be used as an independent prognostic biomarker because this has not been investigated in multivariate analyses. Several CK2 inhibitors have been developed, but CX-4945/silmitasertib is best characterized. This drug has antiproliferative and proapoptotic effects in primary human AML cells. The preliminary results from studies of silmitasertib in the treatment of other malignancies suggest that gastrointestinal and bone marrow toxicities are relatively common. However, clinical AML studies are not available. Taken together, the available experimental and clinical evidence suggests that the possible use of CK2 inhibition in the treatment of AML should be further investigated.

## 1. Introduction

Acute myeloid leukemia (AML) is a heterogeneous and aggressive malignancy characterized by proliferation of transformed immature hematopoietic cells in the bone marrow [1,2]. The overall long-term AML-free survival is only 40–50% even for younger patients who can receive the most intensive chemotherapy possibly combined with allogeneic stem cell transplantation [2]. However, the large group of elderly and unfit patients cannot receive the most intensive treatment; instead, many of them receive only AML-stabilizing therapy and survive for less than a year. Thus, there is a need for new therapeutic strategies both to increase the efficiency of intensive treatment and to prolong the survival for elderly/unfit patients who receive stabilizing treatment.

The protein kinase casein kinase 2 (CK2) is a serine/threonine kinase formed by two catalytic subunits (α or α’; encoded by the genes *CSNK2A1* and *CSNK2A2*, respectively) and one regulatory β subunit (encoded by the *CSNK2B* gene) [3,4]. These units assemble in a tetramer consisting of each of the two α subunits together with two β subunits [3,4]. The β subunit stabilizes the enzyme and is important for the selection of substrates, but the enzyme is constitutively activated and catalytically competent also in the monomeric α form [3]. CK2 seems to have a non-oncogenic action by acting as a growth factor that enhances the oncogenic signaling cascades involved in/important for carcinogenesis, including leukemogenesis [5,6]. Examples of such pathways are nuclear factor-κB (NFκB), Janus kinase/signal transducer and activator of transcription (Jak–Stat), and PI3K–Akt–mTOR; these three pathways are all important in human AML development [3,7,8,9,10,11,12,13,14,15,16,17,18]. This article reviews available studies on the expression and function of CK2 in human AML, as well as the results from experimental AML studies of CK2 inhibition, and we discuss the possible use of CK2 inhibitors in the treatment of human AML.

## 2. An Overview of Important CK2 Effects on Intracellular Signaling

The protein kinase CK2 phosphorylates a wide range of intracellular mediators; it is thereby a regulator of intracellular signaling through several pathways that are important for fundamental cellular functions such as survival, proliferation, migration, protein synthesis, and communication with neighboring cells. Its effects involve the following pathways/functions that are all important in human AML cells:CK2 potentiates Akt (also known as protein kinase B) phosphorylation, and in addition reduces the inhibitory effect of the Akt inhibitor phosphatase and tensin homolog (Pten) on PI3K–Akt–mTOR signaling [17,18]. Increased activity of the PI3K–Akt–mTOR pathway supports cellular survival and proliferation, modulates cellular metabolism, and regulates protein synthesis [3,17,18].CK2 seems to regulate degradation of Ikbα through its C-terminal phosphorylation, increases activation of Ikbα and Ikbβ, and phosphorylates NFκB in the position S529 [15]. This increased signaling through NFκB represents a facilitation of late downstream effects initiated by ligation of several cytokine receptors, and this activation of NFκB activation supports cellular survival and increases the extracellular release of several soluble mediators [3,19,20,21].CK2 regulates the activation of Janus kinase 1 (Jak1) and Jak2 and thereby the activation of their downstream targets Stat1, Stat5, and Stat3; the Stat3 effect is probably mediated at least partly through phosphorylation of S129 [16]. The CK targeting of both Jak2 and Stat3 is important for amplification of downstream signaling initiated by several cytokine receptors [22].DNA repair activity is also altered by CK2 in response to damage signals through its phosphorylation of several substrates, including proteins that are important for repair of double- and single-strand breaks [3,23,24,25].Wnt signaling is also altered by CK2 through effects on downstream targets, including dishevelled proteins (Dv1) and the two downstream mediators β-cathenin and Tcf/Lef (T cell factor/lymphoid enhancer factor) [3]. CK2 (i) activates Dv1 and thereby inhibits degradation of downstream β-cathenin, (ii) phosphorylates and thereby stabilizes β-cathenin, and finally (iii) phosphorylates Tcf/Lef, which forms a complex with β-cathenin that leads to increased transcriptional activity.

As stated above, these five pathways are important for leukemogenesis and/or chemosensitivity in human AML [7,8,9,10,11,12,13,14,26,27,28,29,30]. However, one should emphasize that CK2 only affects certain and relatively few mediators within each of these complex pathways. Each pathway can be regarded as an interacting signaling network regulated by intrapathway positive and negative feedback loops, and individual pathways show interacting crosstalk with other pathways.

The protein kinase CK2 phosphorylates hundreds of physiological substrates and is therefore one of the major contributors to the human phosphoproteome [3]. Thus, the five pathways described in Table 1 represent only a minority of the CK2-targeted proteins, but they represent coordinated effects on key mediators in individual pathways that converge on growth-enhancing and antiapoptotic effects in many cancer cells, including AML cells.

## 3. Importance of the AML Cell Population Used for Experimental and Clinical Studies

### 3.1. AML Cell Populations Have a Hierarchical Organization

AML cell populations have a hierarchical organization; most cells show early spontaneous in vitro apoptosis during culture. The clonogenic cell subset (often <1% of the AML cells) survives and proliferates after at least 7–14 days of culture, whereas a small minority of leukemic stem cells constitutes an AML subset that proliferates in vitro for at least three weeks [34]. Even though one week of in vitro suspension culture seems to enrich the fraction of clonogenic cells [35], leukemic stem cells still constitute a minor subset even after one week of in vitro culture. The total AML cell population has been used in many of the CK2/AML studies that are discussed later in this review.

### 3.2. Methodological Studies for Functional Analysis of AML Stem Cells

Several methodological strategies have been used for evaluation of leukemic stem cells. One strategy is to use long-term in vitro culture for several weeks to enrich stem cells and eliminate other short-lived subsets through in vitro apoptosis; the number of cobblestone colonies after such a long in vitro culture period is regarded as reflecting the number and growth characteristics of the remaining AML stem cells [35]. An alternative strategy is to use xenografting; the AML cell subset with the capacity to recapitulate the disease in xenografted immunocompromised mice is then regarded to include AML stem cells [35]. Finally, the use of flow cytometry to identify the side population and/or CD34^+^CD38^−^ is an alternative to identify an AML cell subset that includes (i.e., is enriched for) leukemic stem cells [36,37,38]. This last methodological strategy has been used in several of the CK2 studies that are discussed later in this review.

A major challenge for the evaluation of stem cell-enriched AML cell populations is that the leukemic stem cell phenotype seems to vary between patients, and for some (exceptional) patients, leukemic stem cells seem to be CD34-negative [39]. Thus, examination of CD34^+^CD38^−^ AML cells as used in many of the CK2 studies discussed below usually includes AML stem cells, but for certain exceptional patients this may not be true.

### 3.3. The Relevance of Investigating the Overall AML Cell Population in Experimental Studies

Even though chemoresistant AML relapse is thought to be derived from AML stem cells [37,40], many experimental studies are still based on the examination of the whole AML cell population or a partial enrichment of a stem cell-containing AML cell subset (e.g., CD34^+^CD38^−^ AML cells). As can be seen by our later discussion, this is also true for studies on CK2.

Even though chemoresistant relapse is an important cause of death in AML and AML relapse is thought to be derived from remaining AML stem cells [1,2,40], it should still be regarded as relevant to investigate the chemosensitivity for the whole AML cell population. First, all AML cells have the same genetic abnormalities that drive leukemogenesis [1,2]. Second, clinical observations have shown that the biological characteristics of the whole AML cell population reflect the risk of later relapse [40]. This is illustrated by the increased risk of relapse for patients having late responses to the initial induction chemotherapy, i.e., detectable morphological disease seven days after treatment induction [40]. Even modulation of the intracellular cell signaling profile of the total AML cell population 24 h post-chemotherapy may be a predictor of long-term patient survival [41]. All of these parameters are based on evaluation of the overall AML cell population. Finally, the gene expression/epigenetic and proteomic profiles in the total AML cell population also reflect clinical chemosensitivity and are associated with AML relapse risk [39,42,43]. Taken together, these observations suggest that the biological characteristics of the overall AML cell population reflect clinically relevant chemosensitivity. One should finally emphasize that the majority of more mature AML cells have a major influence on the overall AML cell microenvironment (e.g., the local cytokine network) in the bone marrow.

Even though the biological characteristics of the overall AML cell population reflect the risk of relapse, an additional relapse-associated characteristic is the distribution of various AML cell subsets within this hierarchically organized cell population. The level of CD34^+^CD38^−^ AML cells is regarded to reflect the burden of leukemic stem cells, and a high burden then seems to be associated with an adverse prognosis in patients receiving intensive AML therapy [37]. Many studies on CK2 in human AML cells are based on examination of the total AML cell population, but as will be described later, studies of enriched CD34^+^CD38^−^ AML cells, an AML stem-including subset, are also available.

## 4. Expression of the Protein Kinase CK2 in Normal Hematopoietic Cells

Previous studies have demonstrated that the level of protein kinase CK2α is relatively low in normal mononuclear (i.e., gradient-separated and T cell-depleted; enrichment of normal immature hematopoietic cells) bone marrow cells derived from healthy individuals [26,44,45]. Thus, for most patients, their CK2 levels are generally higher in AML cells compared to such normal cells.

## 5. The Expression of Protein Kinase CK2 in AML Cells

### 5.1. CK2 Expression in Malignant Cells: Unbalanced Expression and the Expression of Truncated Chains

Studies on various cancer cells have shown that an unbalanced expression of the CK2 α or α’ chains (i.e., monomeric chains not corresponding to CK2β expression) can contribute to malignant transformation [3]. This was first suggested by studies of transformed fibroblasts [46] and later supported by studies of epithelial–mesenchymal transformation [47]. High monomeric levels seem to be associated with an increased risk of metastases in breast cancer and renal cell carcinoma [47,48]. The molecular mechanism behind these effects seems to be a “tumor suppressor-like” function of the CKβ chain, possibly caused by the ability of this chain to prevent/reduce phosphorylation of several mediators that are important for the antiapoptotic effect of CK2 [49]. Thus, this importance of monomeric CK2α has been described for various malignancies and therefore seems to be a more general effect, but it is not known whether this is important for leukemogenesis and/or chemosensitivity in human AML.

Previous studies have also shown that different forms of CK2 chains are present in AML cells [50]. CK2α and CK2α’ have a molecular weight of 44 kDa and 38 kDa, respectively, but CK2α can also be present in a 37 kDa form that seems to be generated by C-terminal proteolysis of the 44 kDa form (i.e., the chain is truncated at its C-terminus) [50]. This truncated form seems to be protected from further degradation. The truncation can be caused both by m-calpain and the 20 S proteasome, but not by the 26 S proteasome. In contrast, the CK2α’ chain seems to be protected from this proteolysis [50]. However, it should be emphasized that the possible importance of the various molecular forms for leukemogenesis and/or chemoresistance in AML has not been investigated.

### 5.2. The Prognostic Impact of CK2 in AML Cells for Patients Receiving Intensive and Potentially Curative Antileukemic Therapy

Two previous studies have investigated the possible prognostic impact of CK2 expression in primary human AML cells. Both of these studies investigated CK2 expression at the protein level, and they both described an association between high intracellular CK2 levels and adverse prognosis.

A study by Kim et al. [44] investigated 48 AML patients with a normal karyotype of their leukemic cells, and the cells were derived at the time of first diagnosis before any antileukemic therapy. The AML cell levels of the CK2α subunit were determined by Western blot analyses, and based on these levels, patients were classified as either high- or low-level patients (i.e., 16 patients having high CK2α levels based on a comparison with α-tubulin expression and a ratio > 0.50; the 32 other patients were defined as having low levels). The protein levels correlated with differences in protein activity. The patients received intensive and thus potentially curative antileukemic therapy, and patients with high CK2α levels had significantly lower AML-free (11.4% vs. 79.9%; *p* = 0.0252) and overall survival (19.3% vs. 64.8%; *p* = 0.0392) estimated after three years.

A study by Aasebø et al. [51] investigated the proteomic and phosphoproteomic profiles of AML cells derived from 41 patients at the time of first diagnosis immediately before induction therapy. All patients received intensive and potentially curative antileukemic therapy, and after an observation period of at least five years, they were classified as either AML-free long-term survivors or patients dying from leukemia relapse. In contrast to a study by Kim at al. [44], these patients represent a consecutive group from a defined geographical area, and as expected, approximately half of the patients had a normal AML cell karyotype. Proteomic and phosphoproteomic comparisons of primary AML cells derived from these two contrasting groups did not show any difference with regard to the CK2α/α’/β chain levels, but increased phosphorylation at several CK2-recognized phosphosites was observed for patients with later relapse. The biological relevance of this phosphoproteomic observation was further supported by functional studies; the protein kinase CK2 inhibitor CX-4945 had a weaker antiproliferative effect for cells derived from relapse patients, i.e., patients with a high degree of phosphorylation at CK2-recognized residues/phosphorylation sites.

Taken together, these two studies suggest that relatively high AML cell levels of CK2, and thereby high levels of phosphorylation at CK2-recognized molecular sites, is a part of an AML cell phenotype associated with clinical chemoresistance/relapse risk.

### 5.3. Possible Associations between CK2-Mediated Phosphorylation Activity and the Antiproliferative Effect of Vacuolar ATPase Inhibitors

A recent study investigated the in vitro antiproliferative effects of vacuolar (V) ATPase inhibitors on primary human AML cells derived from a consecutive group of 80 patients [52]. The antiproliferative effect these inhibitors showed a wide variation between patients. A phosphoproteomic comparison of primary AML cells with strong versus weak/no antiproliferative effects of V-ATPase inhibition identified significant differences in the phosphorylation of several proteins, and several of the molecular sites showing increased phosphorylation for patients with high susceptibility to V-ATPase inhibition had a CK2-recognized motif. This observation suggests that CK2 is important not only for the susceptibility to conventional cytotoxic therapy, but also for the susceptibility to at least certain forms of targeted AML therapy.

## 6. Effects of CK2 Inhibition on AML Cells: Studies of Human Primary Patient Cells and AML Cell Lines

Previous studies have described several hallmarks of cancer that are important for the development and chemosensitivity of human malignancies [53,54]. Several cancer-associated proteins that can be phosphorylated by CK2 are important for these hallmarks [55]. These hallmarks include (i) intracellular cancer cell characteristics (e.g., regulation of apoptosis and proliferation and metabolism, together with genomic instability), (ii) cell surface receptors for and release of components of the extracellular matrix in the cancer cell microenvironment, and (iii) mediators involved in cancer cell communication with neighboring cells in their microenvironment (endothelial cells/angiogenesis, immunomodulation, and invasion/metastasis). The importance of AML cell CK2 levels for various hallmarks of cancer is reviewed and discussed in this section. The effects of CK2 inhibition in human AML cells are summarized in Table 2 [55,56,57,58,59,60,61,62,63,64,65,66,67,68], and they are discussed in more detail in the following subsections.

### 6.1. PI3K–Akt, JAK–STAT, NFκB, and Wnt Signaling, Together with DNA Repair Mechanisms, Are Important Targets for CK2 in Primary Human AML Cells

Studies in primary AML cells have shown that Akt phosphorylation in Ser473 can be detected in both the stem cell-enriched CD34^+^CD38^−^ and the CD34^+^CD38^+^ AML cell subset, and Akt-dependent phosphorylation of Foxo3a in Ser253 has also been detected [56]. Thus, the Akt–Foxo3a pathway is activated, and phosphorylation of NFκB in Ser529 and Stat3 at Ser727 indicates that all three pathways are activated in primary AML cells. However, the phosphorylation/activation of the three pathways is higher for the CD34^+^CD38^+^ AML cell subset than for the stem cell-including CD34^+^CD38^−^ cell subset. The involvement of CK2 in the regulation of these pathways has been further confirmed by CK2 inhibitory studies (see below). Taken together, these observations suggest that the CK2 activity and CK2-mediated activation of signaling pathways vary within the heterogeneous and hierarchically organized AML cell population. Furthermore, Wnt signaling is also important for leukemogenesis/chemosensitivity in human AML [29,30]. Finally, the DNA damage response proteins Xrcc1, Xrcc4, and Mdc1 are also important for leukemogenesis/chemosensitivity in human AML [57,58,59,60], and their phosphorylation is regulated by CK2 [3,55]. Thus, PI3K–Akt, Jak–Stat, NFκB, and Wnt signaling, as well as DNA damage responses, seem to be important substrates for CK2 in human AML cells.

### 6.2. Regulation of Apoptosis in AML Cells: Inhibition of CK2 Has a Proapoptotic Effect

A previous study described a proapoptotic effect of CK2 inhibition in primary human AML cells [26]. This proapoptotic effect of the CX-4945 CK2 inhibitor is dependent on p53, which is regarded as a CK2 target and is important both for DNA repair and regulation of apoptosis [55]. Furthermore, in certain cells, CK2 inhibition causes a secretory senescence phenotype through inhibition of NFκB [21], but it is not known whether this can also occur for AML cells.

The combination of CK2 and PI3K–Akt–mTOR inhibition has a proapoptotic effect in human AML cells [61]. Apigenin was used for CK2 inhibition and LY294002 for PI3K–Akt inhibition in this study, and a proapoptotic effect was seen for the CD34^+^CD38^−^ AML cell population, which is regarded as being enriched of leukemic stem cells. Both inhibitors had proapoptotic effects on primary AML cells, but the combination had a synergistic effect and treatment was associated with decreased CK2 activity and decreased Ser473 Akt phosphorylation. Additional molecular studies have shown that this proapoptotic combination disrupts the mitochondrial membrane potential, induces cytosolic release of cytochrome c and Diablo (two proapoptotic mediators) from mitochondria, and causes caspase-3/8/9 cleavage. At the same time, the levels of several antiapoptotic molecules (Bcl-xl, Mcl-1, Xiap, and survivin) were decreased by the combination. These molecular effects were detected for AML cell lines, primary AML cells, and CD34^+^ AML cells. Finally, this combined treatment decreased Wnt1, β-cathenin, Lrp6 (a coreceptor for Wnt), and Dvl levels in primary AML cells, and Mek1/2 and Erk1/2 phosphorylation were decreased as well as the phosphorylation of p65 NFκB.

The combination of CK2 and PI3K–Akt inhibition possibly represents a dual inhibition that potentiates certain effects of CK2 inhibition because CK2 enhances signaling through this pathway (see Section 2) through phosphorylation of pathway mediators. The combined CK2/PI3K–Akt–mTOR inhibition shows synergistic antileukemic effects that seem to be mediated through complex molecular mechanisms involving altered regulation of apoptosis together with crosstalk between various intracellular signaling pathways (e.g., Mek/Erk). However, it has to be emphasized that these data need to be reproduced because relatively few patient AML cell populations were investigated. Another study described that CK2 inhibition by apigenin alone has a proapoptotic effect, especially in primary AML cells with high CK2α levels [44], and their investigations of AML cell lines also suggest that CK2 inhibition by apigenin causes caspase cleavage, cytosolic release of Diablo, and mitochondrial release of Bax. These last observations further support our assumption (see above) that the effects of combined CK2/PI3K–Akt–mTOR inhibition mainly represent an enhancement of certain CK2 effects.

Apigenin has extensive effects on a wide range of fundamental cellular functions [69,70]; many of these effects seem to be mediated by CK2 inhibition and overlap the effects of other CK2 inhibitors, e.g., CX-4945 [71]. However, apigenin is also a direct inhibitor of topoisomerase 1 [72]; it has effects on lipid metabolism (i.e., the prostaglandin system) that are not seen with CX-54945, an inhibitor [73], and the molecular structure and binding to CK2 are different from other CK2 inhibitors [74]. Thus, apigenin has additional effects that are probably caused by other mechanisms than CK2 inhibitors, and one cannot exclude the possibility that some apigenin effects described in AML studies may be caused by other molecular mechanisms than CK2 inhibition.

### 6.3. The Antiproliferative Versus Proapoptotic Effect of CK2 Inhibition: Effects of CK2 and CK2 Inhibition of Intracellular Signaling

CK2 inhibition of AML cell lines causes an initial (i.e., after 18 h of in vitro exposure) accumulation of cells in S and M phase; this is true for both the CD34^+^CD38^−^ leukemic stem cell-including cell population and the CD34^+^CD38^+^ cell subset [56]. However, later on (after 36 h of exposure), these cells undergo apoptosis and the number of cells in the late phases of the cell cycle decreases. These observations suggest that CK2 is important for regulation of cell cycle progression and the survival/viability in human AML cells.

The heterogeneity of the AML cell population has been further characterized at the mRNA level [56]. A similar overexpression of Foxo3a was then observed for approximately half of the patients, but the Foxo3a target Foxo1 reached detectable expression only for a minority of patients. These observations suggest that patients are heterogeneous with regard to the expression of these two transcriptional regulators in their AML cells. Furthermore, CK2 inhibition caused reduced Akt phosphorylation/activation in Ser473 and reduced Foxo3a phosphorylation in Ser253. The subcellular localization of Foxo3a also seems to be altered by CK2 inhibition with an increased redistribution into the nucleus. Finally, CK2 inhibition caused an expected decrease in NFκB phosphorylation in Ser529 and Stat3 in S727 for both CD34^+^CD38^−^ and CD34^+^CD38^+^ AML cells. Taken together, these observations show that CK2 inhibition can modulate the intracellular functions of all three main CK2 targets, Akt/Stat3/NFκB, in human AML cells both through modulation of phosphorylation/activation and modulation of intracellular compartmentalization for certain pathway proteins.

The polycomb family protein Bmi1 is regarded as important for the maintenance of malignant stem cells [62,63]; it is important in epigenetic regulation and DNA repair. Overexpression (2- to 10-fold increase compared with normal CD34^+^ cells) of this protein at the mRNA level has been described for the CD34^+^CD38^−^ AML cell population for a large majority of patients [56]. CK2 inhibition causes a downregulation of this protein [56]. Thus, CK2 is involved in epigenetic regulation, and this epigenetic modulation is an effect of CK2 inhibition together with histone modulation (see Section 6.4 below).

### 6.4. Altered Epigenetic Regulation in Malignant Diseases: Effects of CK2 Inhibition on Epigenetic Regulation of Gene Expression

Ikaros is a DNA binding protein that regulates target gene transcription through epigenetic modulation; it is a zinc finger transcription factor that regulates the expression of various tumor suppressors/oncogenes and thereby regulates survival and proliferation of leukemic cells [64]. Hyperphosphorylation of Ikaros by CK2 impairs its DNA binding capacity [65]. A recent study described that AML cells showed high baseline expression of phosphorylated Ikaros, CK2 and Bcl-xl, and they further investigated the effect of the CK2 inhibitor CX-4945 on the function of Ikaros in AML [45]. Their study showed that (i) CK2 inhibition decreased Ikaros phosphorylation and Bcl-xl expression, (ii) CK2 overexpression was associated with increased engraftment of xenotransplanted human AML cells, and this engraftment was inhibited by CX-4945 treatment; and (iii) CK2 inhibition increased Ikaros binding to the promoter region of the Bcl-xl gene and Ikaros then functioned as a repressor of Bcl-xl expression. Importantly, this CX-4945-induced repression of antiapoptotic Bcl-xl was mediated by Ikaros and characterized by epigenetic modulation with increased acetylation of lysine 9 (K9) and methylation of lysine 27 of H3 histone proteins. Thus, epigenetic modulations (increased acetylation, decreased methylation, altered nucleosomal aceylation by the SET nuclear proto-oncogene) through posttranscriptional modulation of histones seems to contribute to the proapoptotic effects of CK2 inhibition, and this is also supported by a previous study describing restoration of Ikaros-mediated tumor suppressive activity by CK2 inhibition [45,66,67].

The effect of CK2 inhibition has also been investigated by using a phosphoproteomic strategy for two AML cell lines. These studies described altered phosphorylation/activation of especially proteins involved in chromatin modification and regulation of gene expression, but also for proteins involved in the regulation of cell proliferation, mRNA processing/splicing, ribosome biogenesis, response to DNA damage and protein SUMOylation [68]. Thus, the effect of CK2 inhibition on epigenetic regulation is only one of several effects affecting various hallmarks of cancer.

### 6.5. Hallmarks of Cancer: The Possible Importance of CK2 Phosphorylated Proteins in AML Cells

CK2 has the capacity to phosphorylate a wide range of proteins that are important for various hallmarks of cancer [55]. In Table 3, we indicate how various proteins known to be phosphorylated by CK in human AML would be expected to influence various hallmarks of cancer. An extensive list of such cancer-associated CK2 substrates has been published previously [55]. This summarizing table presents a list of those cancer-associated proteins that have been identified as important CK2 targets in AML, and how we would expect them to influence various hallmarks of cancer.

## 7. Effects of CK2 Inhibition on Intracellular Signaling, Extracellular Molecule, and Cellular Interaction Networks in Human AML: Toll-like Receptor 4 (TLR4) and CK2 Have NFκB as a Common Downstream Target and Thereby Affect AML Stem Cells

As can be seen from our previous discussion, AML cell proliferation and survival is regulated by several interacting networks (Figure 1). First, an intracellular network of cross-communicating signaling pathways with intra-pathway feedback loops is important; several of the AML-associated genetic abnormalities encode proteins that are parts of this pathway network [1,2,7,8,9,10,11,12,13,14]. Second, extracellular molecular networks include the cytokine network that form autocrine and paracrine AML regulatory loops, but several other molecules (e.g., also extracellular matrix molecules, proteases, and protease inhibitors) are involved in this network regulation [35,51,75,76,77]. Third, the network of communicating cells in the bone marrow microenvironment consist of AML cells together with non-leukemic stromal cells and infiltrating immunocompetent cells [34,40]. CK2 is a regulatory molecule at all three levels of networks [56,61], and the function of CK2 as a regulator of Toll-like receptor 4 (TLR4)–NFκB pathway signaling can be used as an example of how CK2 inhibition influences AML cells through effects on all three levels of networks [75,76,77,78,79,80,81,82,83,84,85,86,87,88,89,90,91,92,93,94,95,96,97,98,99,100,101]. The various network effects of TLR4 and its CK2-regulated downstream target NFκB summarized in Figure 1:TLR4 is expressed by AML cells and regulates AML cell proliferation and communication with non-leukemic neighboring cells [9,10,11,12,13,14,16,22,29,30,78,99]. The CK2-regulated NFκB is an important downstream mediator for these TLR4 effects, and NFκB is even regarded as a possible therapeutic target in AML [10,11].TLR4 can initiate downstream signaling by binding of endogenous ligands [76]. Both AML cells and stromal cells release several extracellular matrix (ECM) molecules, and several ECM (derived) molecules can function as endogenous TLR4 ligands and thereby influence the function of the downstream CK2 target NFκB [76].TLR4/NFκB signaling is also important for the phenotypic regulation/modulation several nonleukemic AML-supporting stromal cells, including endothelial cells [85,86,87,88,89,90,91], mesenchymal stem cells (MSCs) [80,92,93], osteoblasts [76,94,95], and immunocompetent cells (e.g., monocytes) [96,97,98,99,100]. One would expect CK2 modulation of NFκB to influence the phenotype/function of these non-leukemic cells.Leukemic and nonleukemic cells communicate through the local cytokine network through their constitutive cytokine release that forms autocrine and paracrine regulatory loops for leukemic and non-leukemic cells, and the CK2-targeted NFκB is a regulator of cellular cytokine release [79,80,81,82,83].The non-hematopoietic bone marrow cells form stem cell niches [97] that support the proliferation and maintenance of normal and probably also malignant stem cells [34]. These niches are formed by several elements, including MSCs, endothelial cells, cells of the osteoblastic lineage, and monocytes/macrophages [97], and in vitro studies have demonstrated that several of these niche-forming cells communicate with and support AML cell proliferation (Figure 1).

The observations described above are mainly based on studies of the overall AML cell population, but several characteristics of the total hierarchically organized AML cell population are shared with AML stem cells (see Section 3), which are thought to be responsible for AML relapse in patients with chemoresistant disease [40].

To summarize, these data support the hypothesis that CK2 targeting has direct effects on the intracellular pathway network of AML (stem) cells, but CK2 inhibition is also expected to affect AML cells indirectly through effects on various non-leukemic neighboring cells in the AML cell microenvironment and their communication with leukemic cells. This is probably true also for the interactions between AML stem cells and the non-leukemic cells in the stem cell niches of the bone marrow.

## 8. Combination of Protein Kinase CK2 Inhibition and Other Antileukemic Therapeutic Strategies

The best characterized CK2 inhibitor is CX-4945/silmitasertib, which can be administered orally and is in early clinical studies [3]. Most of the studies referred to below are based on the use of CX-4549, but other inhibitors have also been developed [102].

### 8.1. Combination of PK2 Inhibition with Conventional Cytotoxic Drugs

A previous study investigated the effect of combining daunorubicin with CX-4945/silmitasertib [26]. These authors described a synergistic antiproliferative effect of these two drugs, and PK2 inhibition reduced daunorubicin-induced STAT3 activation. CK2 inhibition seems to inhibit this daunorubicin-induced antiapoptotic (i.e., potential resistance-inducing) Stat3 activation through downregulation of antiapoptotic Mcl1.

### 8.2. Combination of PK2 Inhibition and the Bcl2 Inhibitor Venetoclax or the Bruton’s Tyrosine Kinase Inhibitor Ibrutinib

The Bcl2-inhibitor venetoclax is used in the treatment of various malignancies, including AML, and various lymphoid malignancies [103,104,105,106,107,108,109]. A recent study identified CK2 as a major regulator of resistance to venetoclax in mantle cell lymphoma; additional experimental studies have demonstrated that even though the CK2 inhibitor CX-4945/silmitasertib alone does not affect the viability of mantle cell lymphoma cell lines or primary samples, CK2 inhibition strongly synergizes with the antilymphoma effects of venetoclax and the Bruton’s tyrosine kinase inhibitor ibrutinib [106,107,108]. This in vitro effect is caused by downregulation of MCL1 levels due to inhibition of the eIF4F complex assembly (i.e., decreased Mcl-1 translation), whereas the levels of Bcl2 and Bcl-xl are not affected. This sensitizing effect thus seems to be due to increased dependency on Bcl2 for cell survival. Furthermore, CK2 targeting also reduces venetoclax resistance mediated by MSCs in coculture experiments. Thus, the anticancer effects of CK2 inhibition in mantle cell lymphoma are probably also mediated through effects on various biological networks at different levels (see Section 7 and Figure 1). CK2 inhibition increases the susceptibility to venetoclax in mantle cell lymphoma through both direct effects on malignant cells (intracellular pathway networks) and through indirect effects on lymphoma-supporting MSCs (cellular network).

Venetoclax has direct antileukemic effects on AML cells and is now used in the treatment of AML [103,104,105] and mantle cell lymphoma [106,107,108,109], and MSCs support the proliferation and survival of both AML cells and mantle cell lymphoma cells [92,93,106,107,108,109,110]. Furthermore, CK2 inhibition causes a similar modulation of intracellular proapoptotic signaling including Mcl1 reduction in AML cells and mantle cell lymphoma cells [61,65,106,107,108,109]. To the best of our knowledge, no studies of combining venetoclax and CK2 inhibition in AML are available, but these similarities between mantle cell lymphoma and AML with regard to CK2 inhibitor effects suggest that this combination should be investigated in AML to clarify whether CK2 targeting also increases AML cell susceptibility to venetoclax.

### 8.3. Combination of CK2 and PI3K–Akt–mTOR Inhibition

As described above, the combination of CK2 and PI3K/Akt inhibition modulates intracellular signaling in AML cells (Section 6.2) [61]. Such combined targeting causes increased apoptosis in AML cells, whereas the effect on normal CD34^+^ bone marrow cells is weaker. Thus, such combined targeting as a possible antileukemic strategy should be further investigated.

### 8.4. Combination of CK2 Inhibition and NFκB and Stat3 Targeting

Static3 is a STAT3 inhibitor that inhibits activation, dimerization, and nuclear translocation of Stat3 [111]. The combination of Static3 and the CK2 inhibitor CX-4945 increases AML cell apoptosis for AML cell lines and primary AML cells, and a similar increased apoptosis has also been observed when CX-4945 is combined with the NFκB inhibitor Bay 11-7082 [61]. These in vitro observations suggest that such combinations of molecular targeting should be further investigated in AML.

## 9. The Toxicity of CK2 Protein Kinase Targeting: The Experience from Experimental and Early Clinical Studies of CX-4945/Silmitasertib in Cancer Patients

### 9.1. In Vitro Studies of the Effects of CK2 Inhibition on Normal Hematopoietic Cells

An experimental study investigated the effect of CK2 inhibition alone and combination of CK2 and PI3K–Akt inhibition on CD34^+^ mononuclear bone marrow cells derived from healthy individuals. When testing low levels of these two inhibitors, the combined treatment had only minor effects on normal hematopoietic cell viability, but a stronger synergistic proapoptotic effect was observed for primary AML cells [61]. The proapoptotic effect on primary AML cells were associated with activation of caspase cascades, disruption of mitochondrial membrane potential, and downregulation of proapoptotic proteins (including Bcl-xl) and NFκB, but these effects were not seen in normal hematopoietic cells. A weaker effect of CK2 inhibition in normal hematopoietic cells than in AML cells was also observed in another study [56].

### 9.2. The General Toxicity of Silmitasertib: The Experience from Early Clinical Studies

CK2 targeting as an anticancer strategy has now been investigated in several early clinical studies. The results from two phase I clinical studies of oral CX-4945/silmitasertib treatment in 43 patients with advanced solid tumors have been reported [112]. These patients received the drug either twice daily or four times daily for the first three consecutive weeks of a four-week cycle, and this treatment was continued until intolerance or disease progression. Diarrhea and hypokalemia were the dose-limiting toxicities, but these toxicities were reversible with discontinuation and/or could be handled by antidiarrheal drugs or potassium supplementation. Furthermore, evidence of biomarker responses (i.e., inhibition of CK2 activity and Akt signaling) could be detected. Thus, oral CX-4945/silmitasertib is safe and the in vivo levels reached by the maximal tolerated dose are sufficient to inhibit intracellular CK2 effects. Hematological toxicity is usually dose-limiting in AML therapy [2], but this was not reported in the preliminary report from these two studies.

In vitro studies suggest that CX-4945/silmitasertib is effective in the treatment of cholangiocarcinoma [113], and the results from a clinical study comparing the combination gemcitabine plus cisplatin with and without oral silmitasertib were recently published [114]. A total of 117 patients were studied, and the treatment toxicity was reported for 30 patients receiving gemcitabine/cisplatin alone and 87 patients receiving gemcitabine/cisplatin together with silmitasertib. The most common treatment-emergent non-hematological adverse grade 3/4 events were diarrhea (70% of patients receiving silmitasertib vs. 13% of the controls), nausea (59% vs. 30%), fatigue (47% in both groups), vomiting (39% vs. 7%), anemia (39% vs. 30%), decreased appetite (31% vs. 30%), hypokalemia (20% vs. 10%), asthenia (20% vs. 3%), cough (20% vs. 7%), and headache (20% vs. none). Furthermore, 15 of the 87 patients experienced severe treatment-emergent adverse events (a total of 27 events) considered to be related to silmitasertib, including anemia, thrombocytopenia and vomiting (three of each), neutropenia, and diarrhea and nausea (one of each). Finally, two deaths in the silmitasertib group experienced fatal toxicity due to septic shock and hepatic failure, respectively. Thus, gastrointestinal toxicity seems to be more common for patients receiving silmitasertib, and such toxicity is also common for patients receiving intensive anti-AML chemotherapy [2].

### 9.3. Hematological and Immunological Side Effects of Silmitasertib: The Experience from the First Clinical Studies

The initial phase I studies of silmitasertib monotherapy did not report hematological toxicity, but in a recent study of silmitasertib used in combination therapy both anemia, neutropenia and thrombocytopenia were reported. This hematological toxicity is presented more in detail in Table 4. It can be seen that hematological toxicity (including severe toxicity) was observed for a relatively large subset of patients, and the frequency of hematological toxicity seemed to be generally higher for patients receiving silmitasertib. The authors only described the side effects without statistical analyses/comparisons, and it should be emphasized that a statistical analysis of differences between silmitasertib patients and controls based on the numbers presented in Table 4 showed a significant difference only for Grade 3/4 thrombocytopenia (Fisher’s exact test, *p*-value 0.02 without correction for the number of comparisons).

AML patients usually have signs of leukemia-induced bone marrow failure at the time of diagnosis, and especially the intensive and potentially curative therapeutic strategies cause treatment-induced bone marrow failure [2]. Studies in an animal CK2α knock-out model suggest that CK2α does not affect the development of myeloid cells (neutrophils and monocytes/macrophages) [115]. Several experimental in vitro studies have shown that normal hematopoietic CD34^+^ cells have low CK2 levels and are less susceptible to CK2 inhibition than AML cells (Section 4, Section 6.2, Section 8.3 and Section 9.1). However, despite these observations, the first clinical studies suggest that there is a risk of even severe hematological toxicity (especially thrombocytopenia) for cancer patients receiving silmitasertib combination therapy.

The effects of CK2 and CK2 inhibition on immunocompetent cells have been summarized in a recent review [96]. First, CK2 seems to modulate the function of neutrophils, and CK2 activation in monocytes/macrophages seems to induce proinflammatory responses. Second, CK2 also seems to be a regulator of dendritic cell functions. Third, experimental animal models suggest that CK2α promotes CD4^+^ T cell proliferation and Th17 cell responses. Finally, CK2α seems important for B cell development and differentiation. Even though it should be emphasized that additional studies of the possible importance of CK2 for innate and adaptive immunity are needed, even these initial studies clearly illustrate the importance of evaluating the risk of hematological and immunological toxicity, as well as the risk of severe infections (e.g., bacterial and fungal infections and viral reactivation) in cancer patients receiving silmitasertib. This is of particular importance for AML patients who often develop severe bone marrow failure following conventional antileukemic therapy.

The CX-4945/silmitasertib inhibitor is not a specific CK2 inhibitor; this agent can also inhibit dual-specificity tyrosine phosphorylation-regulated kinase 1a (Dyrk1a) and glycogen synthase kinase-3beta (Bsk3β) [116]. Both of these molecules can function as regulators of normal myelopoiesis/hematopoiesis and lymphopoiesis [117,118,119,120,121], and these regulatory functions are also operative in vivo [122]. Furthermore, CK2 knock-out does not seem to alter murine hematopoiesis/myelopoiesis. Thus, the hematological toxicity seen during CX-4945/silmitasertib therapy may therefore at least partly be caused by off-target effects. These off-target effects may either be direct effects on hematopoietic cells or alternatively indirect effects via endothelial cells in the stem cell niches [123]. Futures studies of more specific CK2 inhibitors have to clarify these possibilities.

## 10. General Discussion: The Need for Further Studies of Silmitasertib in AML

Most available studies of CK2 in AML cells are based on examination of the total AML cell population, but studies of the AML stem cell-including CD34^+^CD38^−^ cell subset are available [41,56]. However, no studies on CK2 functions/levels or the effects of CK2 inhibition/downregulation using AML cell assays are available. Such studies using long-term culture assays are definitely needed, but an alternative strategy could be to study xenografting models that recapitulate AML development. If xenografting is used, the best methodological strategy would then be to use patient-derived xenografts [96,124,125].

Several studies have described an association between adverse prognosis and high AML cell levels of CK2 [44,51], and these observations suggest that high CK2 levels are part of a chemoresistant AML cell phenotype. This prognostic association is probably caused by the growth-enhancing and/or antiapoptotic effects of CK2 on AML cells, including AML stem cells that are thought to be responsible for relapse [40]. Furthermore, a previous study also described CK2α levels has having no significant associations with age, gender, circulating blast level, morphological classification/differentiation, or complete remission rate [44]. Based on the available data, it is not possible to judge whether the CK2 levels of AML cells should be used in routine clinical practice as a prognostic parameter for AML patients receiving intensive antileukemic treatment, because CK2 levels have not been included in multivariate analyses of prognostic parameters in any clinical studies of human AML.

CK2 is an activator/regulator of both PI3K–Akt and NFκB signaling, but specific inhibition of these CK2-regulated pathways seems to have an additional inhibitory effect when combined with CK2 inhibitors [61]. One possible explanation for this additional effect could be that the inhibition of these pathways caused by CK2 is not complete. An alternative explanation could be that the effects on signaling networks are important, and specific targeting of these pathways represents an additional antileukemic modulation of the signaling networks.

The possibilities of combining CK2 inhibition with other antileukemic treatments are discussed in Section 8. Recent experimental studies of the effects of CK2 suggest that other combined strategies may also become relevant, e.g., combination with bromodomain inhibitors or other epigenetic strategies [126,127,128,129], targeting of AML cell metabolism [130,131], or targeting of CK2 regulators [132].

The CD34^+^CD38^−^ AML cell subset represents a subset that includes immature and undifferentiated leukemic cells, and a high burden of these cells is associated with chemoresistance and increased relapse risk after potentially curative antileukemic treatment [40]. CD34^+^CD38^−^ AML cell variants are often associated with high peripheral blood and bone marrow leukemic cell counts and adverse genetic abnormalities (e.g., complex karyotype and FLT3 mutations). AML stem cells are thought to be responsible for AML relapse after intensive therapy [40]. The CD34^+^CD38^−^ AML cells seem to include leukemic stem cells, and the CD34^+^CD38^−^ AML cell burden may therefore also reflect the AML stem cell burden; this may be the reason for the prognostic impact of the CD34^+^CD38^−^ AML cell level. Several studies have demonstrated that high pretherapy level (i.e., the level at the first time of diagnosis) [133,134] and increased residual levels [133,135,136,137] (especially after two induction cycles [37]) of CD34^+^CD38^−^ AML cells are associated with a higher risk of relapse, shorter remission duration, and/or decreased overall survival. This prognostic impact of the CD34^+^CD38^–^ AML cell burden seems to be independent of white blood cell count and genetic abnormalities [34]. Furthermore, AML stem cell assays are complex and time-consuming and not suitable for routine practice, whereas estimation of the CD34^+^CD38^−^ AML cell burden is not a time-consuming analysis and the results can therefore be included in the initial/pre-chemotherapy patient evaluation. This prognostic impact also suggests that more specific targeting of this particular AML cell subset should be tried in AML therapy; CK2 inhibition may then be a possible strategy because CK2 inhibition has antiproliferative and pro-apoptotic effects in CD34^+^CD38^−^ AML cells [56,61].

## 11. Conclusions

High AML cell levels of CK2 seem to be a part of an aggressive AML cell phenotype, and experimental studies have demonstrated that CK2 inhibition has antileukemic effects in human AML cells. We therefore conclude that CK2 inhibition should be further investigated as a possible antileukemic strategy in human AML, but future clinical studies have to carefully evaluate its toxicity, especially bone marrow toxicity, that often is dose-limiting in AML treatment.

## Figures and Tables

**Figure 1 cancers-15-03711-f001:**
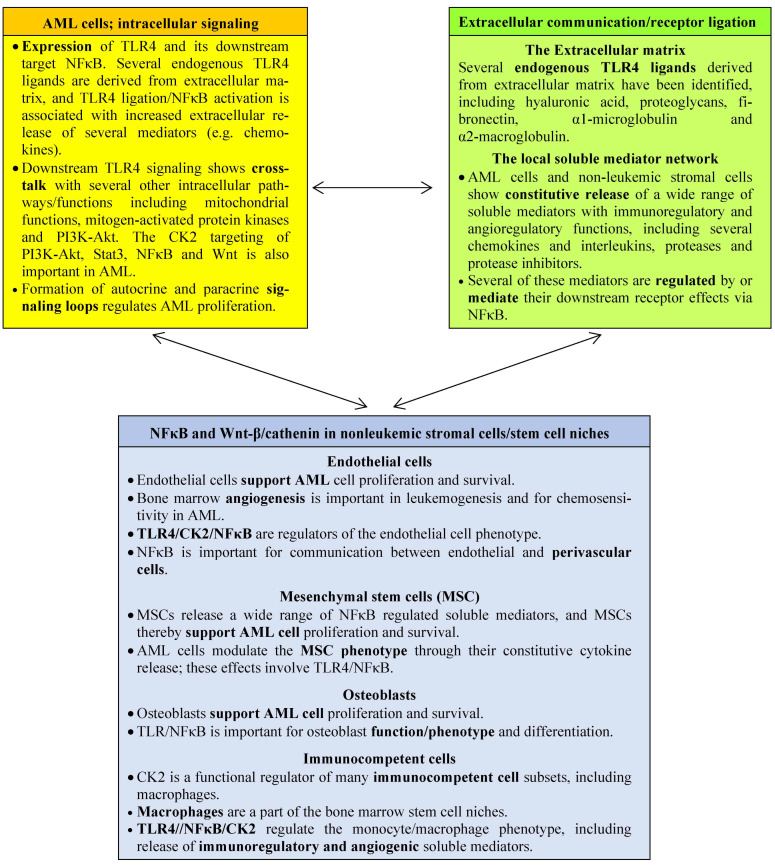
Targeting of complex biological networks in AML by CK2 inhibition; TLR4/CK2/NFκB interactions as an example of the complexity of CK2 targeting. NFκB is a common target for downstream TLR4 signaling and CK2-mediated phosphorylation; leukemogenesis/chemosensitivity is thereby modulated through three interacting networks: (i) Cross-talking intracellular signaling pathways (yellow) [9,12,13,14,16,22,71,75,76,77,78], (ii) the local extracellular microenvironment (extracellular matrix and cytokines; green) [76,79,80,81,82,83], and (iii) the cellular network with interactions between the leukemic cells and their neighboring non-leukemic endothelial cells [84,86,87,88,89,90,91], MSC [80,92,93], osteoblasts [76,94,95] and immunocompetent cells [96,97,98,99,100,101] (all in blue). Thus, the effects of TLR4/NFκB intracellular signaling are modulated by CK2-mediated phosphorylation of the common target NFκB.

**Table 1 cancers-15-03711-t001:** CK2 regulated intracellular signaling pathways that are important for leukemogenesis and/or chemosensitivity in primary human AML cells [3,15,16,17,18,19,20,21,22,23,24,25,26,31,32,33]. The effects on five different pathways are summarized, and for each pathway we present CK2-targeted mediators (left column/target), the CK2 effect on the targeted pathway mediator (middle column), and the cellular consequences of CK2-mediated mediator phosphorylation (right column).

Target	Phosphorylation	Cellular Consequences of Phosphorylation
**Regulation of PI3K–Akt–mTOR signaling**		
Akt	Phosphorylation including Ser129	Ser129 phosphorylation prevents dephosphorylation of Thr308; this leads leading to downstream mTOR signaling
Pten	Phosphorylation of Ser370, as well as several other residues	Inhibits its phosphatase activity, thereby preventing downregulation of PI3K-dependent signaling. Pten is destabilized both by this Ser370 phosphorylation and phosphorylation of Thr366 by glycogen synthase kinase 3
**Regulation of NFκB signaling**		
Ikk	Ser32 and Ser36 phosphorylation	Ikk activation with downstream phosphorylation of IκBα
IκBα	Increased C-terminal phosphorylation through a direct CK2 effect and indirectly via Ikk	Phosphrylation of this inhibitor promotes its degradation, leading to NFκBp65 translocation to the nucleus
NFκBp65	Ser529 phosphorylation	Increased transcriptional activity
**Jak2/Stat3 signaling**		
Jak1	Activated by phosphorylation	Increased survival and proliferation, modulated cytokine release, and thereby altered communication with neighboring non-leukemic and AML-supporting cells
Jak2	Activated by phosphorylation	
Stat3	Activated by phosphorylation	
**DNA damage responses**		
Xrcc4	The general effect of CK2-mediated phosphorylation is an increased association with DNA–repair protein complexes	Double strand DNA repair
Mre11		Involved in several mechanisms of DNA repair, incuding double-strand repair
Xrcc1		Single-strand DNA repair
Mdc1		A scaffold protein involved in the early steps of DNA repair
**Wnt/β-cathenin signaling**		
Dvl	Multisite modulation of the signaling pathway through phosphorylation of different mediators	Reducing β-cathenin degradation
β-cathenin		Increased nuclear translocation and transcriptional ctivity
TCF/LEF		Increased transcription factor activity

Abbreviations: Dvl, dishevelled proteins; mTOR, molecular target of rapamycin; Pdk1, phosphoinositide-dependent kinase 1; Pten, phosphatase and tensin homolog.

**Table 2 cancers-15-03711-t002:** A summary of the important molecular and functional effects of CK2 inhibition in AML cells; an overview of the molecular effects and modulation of organellar functions [55,56,57,58,59,60,61,62,63,64,65,66,67,68]. The effects are discussed in more detail in Section 6.1, Section 6.2, Section 6.3, Section 6.4 and Section 6.5.

Pathway/Function	Molecules/Mechanism
Regulators of apoptosis	Increased proapoptotic effect of p53 Disrupted mitochondrial potential with increased release of cytochrome c and Diablo to cytosol Cleavage of caspases 3/8/9 Decreased levels of proapoptotic Bcl-xl, Mcl-1, Xiap, and survivin
Intracellular signaling	Decreased Akt phosphorylation and decreased levels of Wnt1, β-cathenin, the Lrp6 coreceptor, and Dvl Decreased p65, Mek1, and Erk1/2 phosphorylation
Cell cycle	Accumulation of cells in S and M phase
Transcriptional regulation	Reduced phosphorylation and subcellular localization of Foxo3a with increased redistribution to the nucleus Decreased NFκB, Stat3, and Ikaros phosphorylation
Epigenetic regulation and DNA repair	Increased activity of p53 Downregulation of the epigenetic regulator Bmi1 Histone modulation (methylation/acetylation) by Ikaros

**Table 3 cancers-15-03711-t003:** The influence of important CK2 molecular targets on various hallmarks of cancer; a summary of the CK2 substrates in intracellular pathways (P3K–Akt, JAK–STAT, NFκB, DNA repair, Wnt) known to be important for leukemogenesis and chemosensitivity in primary human AML cells. The classification of individual mediators is based on reference [55].

Hallmark of Cancer	Phosphorylated Molecule/Control of Mediator Release
Regulation of apoptosis in AML cells	Akt, β-cathenin, p53, Pten
Proliferation of AML cells	Akt, Erk, IκBβ, NFκB, p65, Stat1, Stat3, Stat5, β-cathenin, Pten
Genome instability/DNA repair	Mre11, Xrcc1, β-cathenin, Xrcc4
Metabolic regulation of malignant cells	Akt–mTORC2, β-cathenin, mTORC1, Pten
Local malignant cell (bone marrow) infiltration	β-cathenin
Interactions with the tumor microenvironment Modulation of immunocompetent cells Local bone marrow angiogenesis	Akt, β-cathenin, fibronectin NFκB controls the release of a wide range of soluble mediators, including chemokines that are involved in leukocyte chemotaxis and local angiogenesis

**Table 4 cancers-15-03711-t004:** Hematological toxicity for cholangiocarcinoma patients receiving either gemcitabine/cisplatin alone (30 patients, referred to as controls) or in combination with silmitasertib (87 patients, referred to as Silmitasertib). The results are presented as the number of patients with the percentage of patients given in parentheses [114].

Toxicity	Anemia		Thrombocytopenia		Neutropenia	
	Silmitasertib	Controls	Silmitasertib	Controls	Silmitasertib	Controls
All grades	34 (39%)	9 (30%)	24 (28%)	2 (7%)	27 (31%)	7 (23%)
Grade ≥ 3	17 (20%)	5 (17%)	11 (12%)	1 (3%)	20 (23%)	6 (20%)

## Data Availability

Not applicable.

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
