# Peer review of "Casein Kinase 2 (CK2): A Possible Therapeutic Target in Acute Myeloid Leukemia"

_cancers, 2023, doi:10.3390/cancers15143711_

Round 1
Reviewer 1 Report
This is a thorough review by the authors who have an outstanding research record in studying signaling pathways in hematopoietic malignancies and are world exerts in the acute myeloid leukemia (AML). The authors provide a comprehensive review of the role of Casein Kinase II (CK2) in regulation of AML cells. The first part of the manuscript provides an extensive overview of the role of CK2 in regulation of various cellular function and signaling pathways in normal and malignant AML cells. The second part of the manuscript reviews the effect of the upregulation of CK2 in AML and AML stem cells. The third part of the manuscript reviews the use of CK2 inhibition, with a special emphasis on CX-4945 (Silmitasertib), as a single drug or and/or in combination with other chemotherapy for treatment of AML. Toxicity of CK2 inhibitors are reviewed in the last part of the manuscript.
This is an outstanding review by leaders in the AML and signal transduction field that covers in detail an important topic that is of high general interest for a broad clinical and scientific audience. The strongest aspect of this review is that focuses on the molecular mechanisms through which CK2 regulates signaling pathways in AML, but also reviews the translational and clinical significance of CK2 inhibitors for treatment of AML.
There is one minor concern, which should be addressed in order for the manuscript to be acceptable for publication: In lines 561-570 of the manuscript, the authors discuss hematological toxicity of CK2 inhibitor, Silmitasertib, which was observed in Phase I clinical trials, along with lack of hematological toxicity in mouse CK2 alpha knockout model. It would be helpful for the broad audience of Cancers, to mention that a possible hematological toxicity of Silmitasertib is potentially (if not likely) result of off-target effect of this drug – it is known that Silmitasertib can also inhibit other kinases besides CK2 (e.g. DYRK1A)
In summary, this is an outstanding manuscript that includes a review of the role of CK2 in AML, along with clinical impact of CK2 inhibition in AML. The review covers an important topic and this manuscript is very suitable for publication in Cancers.
Author Response
1.1.There is one minor concern, which should be addressed in order for the manuscript to be acceptable for publication: In lines 561-570 of the manuscript, the authors discuss hematological toxicity of CK2 inhibitor, Silmitasertib, which was observed in Phase I clinical trials, along with lack of hematological toxicity in mouse CK2 alpha knockout model. It would be helpful for the broad audience of Cancers, to mention that a possible hematological toxicity of Silmitasertib is potentially (if not likely) result of off-target effect of this drug – it is known that Silmitasertib can also inhibit other kinases besides CK2 (e.g. DYRK1A)
Response: We are very grateful for this comment. A new chapter has been added to Section 7 (the last chapter in the Revised Version), and we have also added seven new references (references 116-123) that are used in this chapter.

Reviewer 2 Report
This is a review paper of Casein Kinase 2 as a therapeutic target in AML. It covers the theme
from basic science to clinical studies comprehensively. This topic should be interesting to
many Hem/Onc doctors. I have only a couple of request and suggestions to the authors.
l Table 1 is indistinct and confusing. Maybe colored row would be helpful to make the table clearer or use a different format.
l P4 L167: What does this sentence “CK2α is relatively low in normal mononuclear” mean? Compared to what? The following sentence does not make sense to me either. Please show the frequency of high CK2α in AML with references.
l P5 L203: CX2α should be CK2α
l P5 L204: It would be clearer if the authors add the number of CK2αhigh and low patients from this reference.
l P12L487 Although both for MCL not for AML, there are at least two publications of combining venetoclax and CK2 inhibitor at this point. (Thus YJ, et al. Haematologica 2023, Manni S et al. Front Cell Dev Biol 2022.) The authors can include these studies to discuss the combination.
Author Response
2.1. Table 1 is indistinct and confusing. Maybe colored row would be helpful to make the table clearer or use a different format.
Response: The Table legend has been rewritten, the text in the table has been modified/simplified and additional abbreviations are explained. We hope our solutions can be accepted.
2.2. P4 L167: What does this sentence “CK2α is relatively low in normal mononuclear” mean? Compared to what? The following sentence does not make sense to me either. Please show the frequency of high CK2α in AML with references.
Response: This sentence has been rewritten (see Section 4).
2.3. P5 L203: CX2α should be CK2α
Response: This has been corrected (Section 5.2 second chapter).
2.4. P5 L204: It would be clearer if the authors add the number of CK2αhigh and low patients from this reference.
Response: This has now been added (Section 5.2 second chapter). We have also included an additional brief comment on associations between CK2α levels and lack of associations between clinical parameters in Section 10 (page 16).
2.5. P12L487 Although both for MCL not for AML, there are at least two publications of combining venetoclax and CK2 inhibitor at this point. (Thus YJ, et al. Haematologica 2023, Manni S et al. Front Cell Dev Biol 2022.) The authors can include these studies to discuss the combination.
Response: These two references have been added (references 107 and 108). They are referred to in Section 8.2 first chapter and in the rewritten Section 8.2. second chapter.
New references:
Manni S, Pesavento M, Spinello Z, Saggin L, Arjomand A, Fregnani A, Quotti Tubi L, Scapinello G, Gurrieri C, Semenzato G, Trentin L, Piazza F. Protein Kinase CK2 represents a new target to boost Ibrutinib and Venetoclax induced cytotoxicity in mantle cell lymphoma. Front Cell Dev Biol. 2022 Aug 11;10:935023. doi: 10.3389/fcell.2022.935023. PMID: 36035991; PMCID: PMC9403710.
Thus YJ, De Rooij MFM, Swier N, Beijersbergen RL, Guikema JEJ, Kersten MJ, Eldering E, Pals ST, Kater AP, Spaargaren M. Inhibition of casein kinase 2 sensitizes mantle cell lymphoma to venetoclax through MCL-1 downregulation. Haematologica. 2023 Mar 1;108(3):797-810. doi: 10.3324/haematol.2022.281668. PMID: 36226498; PMCID: PMC9973496.

Reviewer 3 Report
This is an interesting, well-written, and well-organized review about the specific role of CK2 in AML and the rationale of its targeting in this leukemia. It has the merit of critically evaluating the available studies, by discussing the appropriateness of the methodological approaches applied, as well as the results of the clinical investigations.
I have some minor points, as follows:
1. Table 1:
- PI3K/Akt signaling: no evidence supports a role for CK2 in phosphorylating Akt Ser473, please correct. In addition, the phosphorylation of Akt Thr308 is not due to “Activation and stabilization of Pdk1”, but to the Akt S129 phosphorylation and consequent prevention of T308 dephosphorylation, as demonstrated in Di Maira et al., 2009, doi: 10.1007/s00018-009-0108-1. All the CK2/PI3K-Akt connections are reviewed in Bertacchini et al., 2017 doi: 10.1016/j.jbior.2017.03.002.
On PTEN, CK2 has been reported to phosphorylate several residues, not only S370 (see Fragoso and Barata, 2015, doi: 10.1016/j.ymeth.2014.10.015
- Jak/STAT3 signaling: to the best of the reviewer’s knowledge, CK2 phosphorylates S727 of STAT3, not of Jak1 (see for example ref 26 of the manuscript)
- Dishevelled proteins are usually abbreviated by Dvl, and not Dv, thus Dv1 should be Dvl1. However, since CK2 affects different Dvl isoforms, I suggest just indicating Dvl.
2. The title of Section 3 looks like a method section of an experimental original article. I suggest changing it to something like: “Importance of the AML cell population used for experimental and clinical studies”.
The title of Section 7 is also rather confusing, I suggest simplifying it
3. Indeed, Section 7 is not very clear, in particular, I suggest modifying Figure 1. While I appreciate its organization into the three levels, I did not find it clear what the role of CK2 is on each level, as most sentences do not refer to CK2. I understand that in some cases the effects of CK2 targeting might be indirect, but this should be somehow highlighted in the figure; probably, the message would be more effective and clearer in a more schematic figure.
Moreover, if the TLR4/NFkB is analyzed only as an example (extendable to several other signaling molecules), this should be evident from the very title of the figure
4. When commenting on ref [58] (lines 305-306) it could be also added that apigenin is a very unspecific inhibitor of CK2 (see for example Salehi B et al., 2019, 10.3390/ijms20061305)
5. Ref 93 for CK2 inhibitors is quite old, please update (e.g. Borgo and Ruzzene, 2021, doi: 10.1016/bs.apcsb.2020.09.003.)
6. Line 502: is ref [54] the right one? It doesn’t seem so
7. Table 4: Could the Authors comment on the statistical significance of the different effects observed in treated and control patients?
8. There are several typos, or sentences that need correction. See for example:
- line 17 (syntax)
- Table 2: “surviving” should be “survivin”
- line 344: “og” (of)
- line 373: “CK” (CK2)
- line 399: add “are” before “summarized”
- line 400: delete “of”
- line 409: add “of” before “several...”
- line 415: “targeted” should be “target”
- line 459: CX-4945, not CX-4548
- line 466: CK2, not PK2
- line 595: “proapoptotic” is wrong, change to “anti-apoptotic”
Author Response
3.1. Table 1:
- PI3K/Akt signaling: no evidence supports a role for CK2 in phosphorylating Akt Ser473, please correct. In addition, the phosphorylation of Akt Thr308 is not due to “Activation and stabilization of Pdk1”, but to the Akt S129 phosphorylation and consequent prevention of T308 dephosphorylation, as demonstrated in Di Maira et al., 2009, doi: 10.1007/s00018-009-0108-1. All the CK2/PI3K-Akt connections are reviewed in Bertacchini et al., 2017 doi: 10.1016/j.jbior.2017.03.002.
- On PTEN, CK2 has been reported to phosphorylate several residues, not only S370 (see Fragoso and Barata, 2015, doi: 10.1016/j.ymeth.2014.10.015
- Jak/STAT3 signaling: to the best of the reviewer’s knowledge, CK2 phosphorylates S727 of STAT3, not of Jak1 (see for example ref 26 of the manuscript)
- Dishevelled proteins are usually abbreviated by Dvl, and not Dv, thus Dv1 should be Dvl1. However, since CK2 affects different Dvl isoforms, I suggest just indicating Dvl.
Response: We are very grateful for these correction. The three suggested references have been included (references 31-33). All suggested corrections made the reviewer have incorporated into the table:
- The description of Akt has been rewritten:
- It is stated that CK2 phosphorylates several residues on Pten;
- The description of Jak1 has been corrected;
- The abbreviation Dvl is explained in the footnote and used as suggested by the reviewer.
- The suggested references have been added/included.
3,2. The title of Section 3 looks like a method section of an experimental original article. I suggest changing it to something like: “Importance of the AML cell population used for experimental and clinical studies”.
The title of Section 7 is also rather confusing, I suggest simplifying it
Response: The titles of Sections 3 and 7 have been rewritten as suggested by the reviewer.
3.3. Indeed, Section 7 is not very clear, in particular, I suggest modifying Figure 1. While I appreciate its organization into the three levels, I did not find it clear what the role of CK2 is on each level, as most sentences do not refer to CK2. I understand that in some cases the effects of CK2 targeting might be indirect, but this should be somehow highlighted in the figure; probably, the message would be more effective and clearer in a more schematic figure.
Moreover, if the TLR4/NFkB is analyzed only as an example (extendable to several other signaling molecules), this should be evident from the very title of the figure
Response: We have redesigned the figure, we have simplified the text in the figure, and the figure legend has been rewritten. We hope our solutions can be accepted.
3.4. When commenting on ref [58] (lines 305-306) it could be also added that apigenin is a very unspecific inhibitor of CK2 (see for example Salehi B et al., 2019, 10.3390/ijms20061305).
Response: We agree that the possible off-target effects of CK2 inhibitors are important (see also the comment made by reviewer 1), and we have therefore added a new chapter at the end of Section 6.2 where this is commented. We have also added six new references (references 69-74).
New references:
Salehi B, Venditti A, Sharifi-Rad M, Kręgiel D, Sharifi-Rad J, Durazzo A, Lucarini M, Santini A, Souto EB, Novellino E, Antolak H, Azzini E, Setzer WN, Martins N. The Therapeutic Potential of Apigenin. Int J Mol Sci. 2019 Mar 15;20(6):1305. doi: 10.3390/ijms20061305. PMID: 30875872; PMCID: PMC6472148.
Javed Z, Sadia H, Iqbal MJ, Shamas S, Malik K, Ahmed R, Raza S, Butnariu M, Cruz-Martins N, Sharifi-Rad J. Apigenin role as cell-signaling pathways modulator: implications in cancer prevention and treatment. Cancer Cell Int. 2021 Apr 1;21(1):189. doi: 10.1186/s12935-021-01888-x. PMID: 33794890; PMCID: PMC8017783.
McCarty MF, Assanga SI, Lujan LL. Flavones and flavonols may have clinical potential as CK2 inhibitors in cancer therapy. Med Hypotheses. 2020 Aug;141:109723. doi: 10.1016/j.mehy.2020.109723. Epub 2020 Apr 9. PMID: 32305811.
Fux JE, Lefort ÉC, Rao PPN, Blay J. Apigenin directly interacts with and inhibits topoisomerase 1 to upregulate CD26/DPP4 on colorectal carcinoma cells. Front Pharmacol. 2022 Dec 22;13:1086894. doi: 10.3389/fphar.2022.1086894. PMID: 36618939; PMCID: PMC9815539.
Suhas KS, Parida S, Gokul C, Srivastava V, Prakash E, Chauhan S, Singh TU, Panigrahi M, Telang AG, Mishra SK. Casein kinase 2 inhibition impairs spontaneous and oxytocin-induced contractions in late pregnant mouse uterus. Exp Physiol. 2018 May 1;103(5):621-628. doi: 10.1113/EP086826. Epub 2018 Apr 15. PMID: 29708304.
Sarno S, Moro S, Meggio F, Zagotto G, Dal Ben D, Ghisellini P, Battistutta R, Zanotti G, Pinna LA. Toward the rational design of protein kinase casein kinase-2 inhibitors. Pharmacol Ther. 2002 Feb-Mar;93(2-3):159-68. doi: 10.1016/s0163-7258(02)00185-7. PMID: 12191608.
